# *Straightjacket/α2δ3* deregulation is associated with cardiac conduction defects in myotonic dystrophy type 1

Emilie Auxerre-Plantié[1], Masayuki Nakamori[2], Yoan Renaud[1], Aline Huguet[3,4], Caroline Choquet[5], Cristiana Dondi[1], Lucile Miquerol[5], Masanori P Takahashi[6], Geneviève Gourdon[3,4], Guillaume Junion[1], Teresa Jagla[1], Monika Zmojdzian[1]*, Krzysztof Jagla[1]*

[1]GReD, CNRS UMR6293, INSERM U1103, University of Clermont Auvergne, Clermont-Ferrand, France; [2]Department of Neurology, Osaka University Graduate School of Medicine, Osaka, Japan; [3]Imagine Institute, Inserm UMR1163, Paris, France; [4]Centre de Recherche en Myologie, Inserm UMRS974, Sorbonne Universités, Institut de Myologie, Paris, France; [5]Aix-Marseille Univ, CNRS UMR 7288, IBDM, Marseille, France; [6]Department of Functional Diagnostic Science, Osaka University Graduate School of Medicine, Osaka, Japan

*For correspondence:
monika.zmojdzian@uca.fr (MZ);
christophe.jagla@udamail.fr (KJ)

Competing interests: The authors declare that no competing interests exist.

**Abstract** Cardiac conduction defects decrease life expectancy in myotonic dystrophy type 1 (DM1), a CTG repeat disorder involving misbalance between two RNA binding factors, MBNL1 and CELF1. However, how DM1 condition translates into conduction disorders remains poorly understood. Here we simulated MBNL1 and CELF1 misbalance in the *Drosophila* heart and performed TU-tagging-based RNAseq of cardiac cells. We detected deregulations of several genes controlling cellular calcium levels, including increased expression of straightjacket/α2δ3, which encodes a regulatory subunit of a voltage-gated calcium channel. Straightjacket overexpression in the fly heart leads to asynchronous heartbeat, a hallmark of abnormal conduction, whereas cardiac straightjacket knockdown improves these symptoms in DM1 fly models. We also show that ventricular α2δ3 expression is low in healthy mice and humans, but significantly elevated in ventricular muscles from DM1 patients with conduction defects. These findings suggest that reducing ventricular straightjacket/α2δ3 levels could offer a strategy to prevent conduction defects in DM1.

## Introduction

Myotonic dystrophy type 1 (DM1), the most prevalent muscular dystrophy in adults (*Theadom et al., 2014*), is caused by a CTG triplet repeat expansion in the 3' untranslated region of the *dystrophia myotonica protein kinase* (*dmpk*) gene. Transcripts of the mutated *dmpk*, containing expanded CUGs, form hairpin-like secondary structures in the nuclei, and sequester RNA-binding proteins with affinity to CUG-rich sequences. Among them, Muscleblind-like 1 (MBNL1) protein is trapped within the repeats, forming nuclear *foci* aggregates that hallmark the disease (*Davis et al., 1997*; *Taneja et al., 1995*). In parallel, the CUGBP- and ELAV-like family member 1 (CELF1) is stabilized (*Kuyumcu-Martinez et al., 2007*), creating misbalance between MBNL1 and CELF1. This leads to missplicing of several transcripts and a general shift from adult to fetal isoforms (*Freyermuth et al., 2016*; *Kino et al., 2009*; *Savkur et al., 2001*). In addition, repeat toxicity induces a range of splice-independent alterations including impaired transcript stability (*Sicot et al., 2011*). A combination of splice-dependent and splice-independent events thus underlies DM1 pathogenesis, with the latter remaining largely unexplored. DM1 affects mainly skeletal muscles and the heart, with about 80% of

DM1 patients showing impaired heart function with arrhythmia and conduction disturbance, which can sometimes end in heart block and sudden death (*de Die-Smulders et al., 1998*; *Groh et al., 2008*; *Mathieu et al., 1999*). Cardiac symptoms, and particularly conduction defects, thus decrease life expectancy in DM1 (*Wang et al., 2009*). Data suggest that cardiac phenotypes, including conduction defects, are due to MBNL1/CELF1 misbalance. It was shown in a DM1 mouse model that PKC phosphorylates CELF1 leading to increased CELF1 levels, whereas PKC inhibition caused CELF1 reduction and amelioration of cardiac dysfunction (*Wang et al., 2009*). This suggests that increased CELF1 levels could cause heart phenotypes in DM1, a possibility supported by findings that heart-specific upregulation of CELF1 reproduces functional and electrophysiological cardiac changes observed in DM1 patients and mouse model (*Koshelev et al., 2010*). In parallel, analyses of *Mbnl1* mutant mice (*Dixon et al., 2015*) and evidence that misregulation of MBNL1-splice target gene *SCN5A* encoding a cardiac sodium channel leads to cardiac arrhythmia and conduction delay (*Freyermuth et al., 2016*), indicate that Mbnl1 contributes to DM1 heart phenotypes. However, in spite of aberrant SCN5A splicing (*Freyermuth et al., 2016*) and downregulation of a large set of miRNAs (*Kalsotra et al., 2014*), gene deregulations causing cardiac dysfunctions in DM1 remain to be characterized.

To gain further insight into mechanisms underlying cardiac DM1 phenotypes, we used previously described *Drosophila* DM1 models (*Picchio et al., 2013*). The heart of the fruit fly is simple in structure, but like the human heart, it displays pacemaker-regulated rhythmic beating, involving functions of conserved ion channels (*Ocorr et al., 2007*; *Taghli-Lamallem et al., 2016*). We simulated pathogenic MBNL1/CELF1 misbalance specifically in the fly heart by attenuating the *Drosophila MBNL1* ortholog *Muscleblind (Mbl)* and by overexpressing the *CELF1* counterpart *Bruno-3 (Bru-3)* (*Picchio et al., 2018*). This caused asynchronous heartbeat (anterior and posterior heart segments beating at different rates), which in *Drosophila* results from partial conduction block (*Birse et al., 2010*). Using these two fly DM1 models, we hoped to identify molecular players involved in DM1-associated conduction defects. We did not observe asynchronous heartbeats in flies expressing in the heart 960CTG repeats. This DM1 model (*Picchio et al., 2013*) developed other cardiac phenotypes such as arrhythmia. To identify deregulated genes underlying conduction defects, we applied a heart-targeted TU-tagging approach (*Miller et al., 2009*) followed by RNA sequencing. This cardiac cell-specific genome-wide approach yielded a discrete number of evolutionarily conserved candidate genes with altered cardiac expression in both DM1 models used, including regulators of cellular calcium. Among them, we found increased transcript levels of *straightjacket* (*stj*)/*CACNA2D3* ($\alpha 2\delta 3$), which encodes a major regulatory subunit of Ca-$\alpha$1D/Ca$_v$1.2 voltage-gated calcium channel, and is a key regulator of calcium current density, which triggers cardiac contraction (*Bodi, 2005*; *Dolphin, 2013*; *Hoppa et al., 2012*; *Mesirca et al., 2015*). The role of *stj* transcript level in proper conduction is supported by cardiac-specific overexpression of *stj*, which leads to asynchronous heartbeat. Conversely, attenuating its expression by cardiac-specific knockdown in both DM1 fly heart models reverses cardiac asynchrony. Our hypothesis that *stj* contributes to the cardiac DM1-associated pathology is supported by our finding that ventricular $\alpha 2\delta 3$ expression level is low in healthy mouse and human hearts, but is significantly increased in DM1 patients with cardiac conduction defects. Hence lowering $\alpha 2\delta 3$ in ventricular cardiomyocytes could offer a potential treatment strategy for DM1-associated conduction defects and in particular intraventricular conduction delay (IVCD).

## Results

### Imbalance of MBNL1 and CELF1 counterparts in the *Drosophila* heart results in asynchronous heartbeat

The *Drosophila* counterparts of MBNL1 and CELF1, the Mbl and Bru-3 proteins, are both present in the fly heart cells, including contractile cardiomyocytes, and localize predominantly but not exclusively to the nuclei (*Figure 1—figure supplement 1*). To mimic the DM1-associated imbalance between these two RNA binding factors, we either knocked down *Mbl*, or overexpressed *Bru-3*. We used the GAL4/UAS system with a cardiac-specific GAL4 driver line, *Hand-GAL4* to target UAS-driven transgene expression in the adult fly heart (*Figure 1A*). We found that the *UAS-MblRNAi* line efficiently attenuated *Mbl* expression, whereas *UAS-Bru-3* generated *Bru-3* overexpression in GAL4-

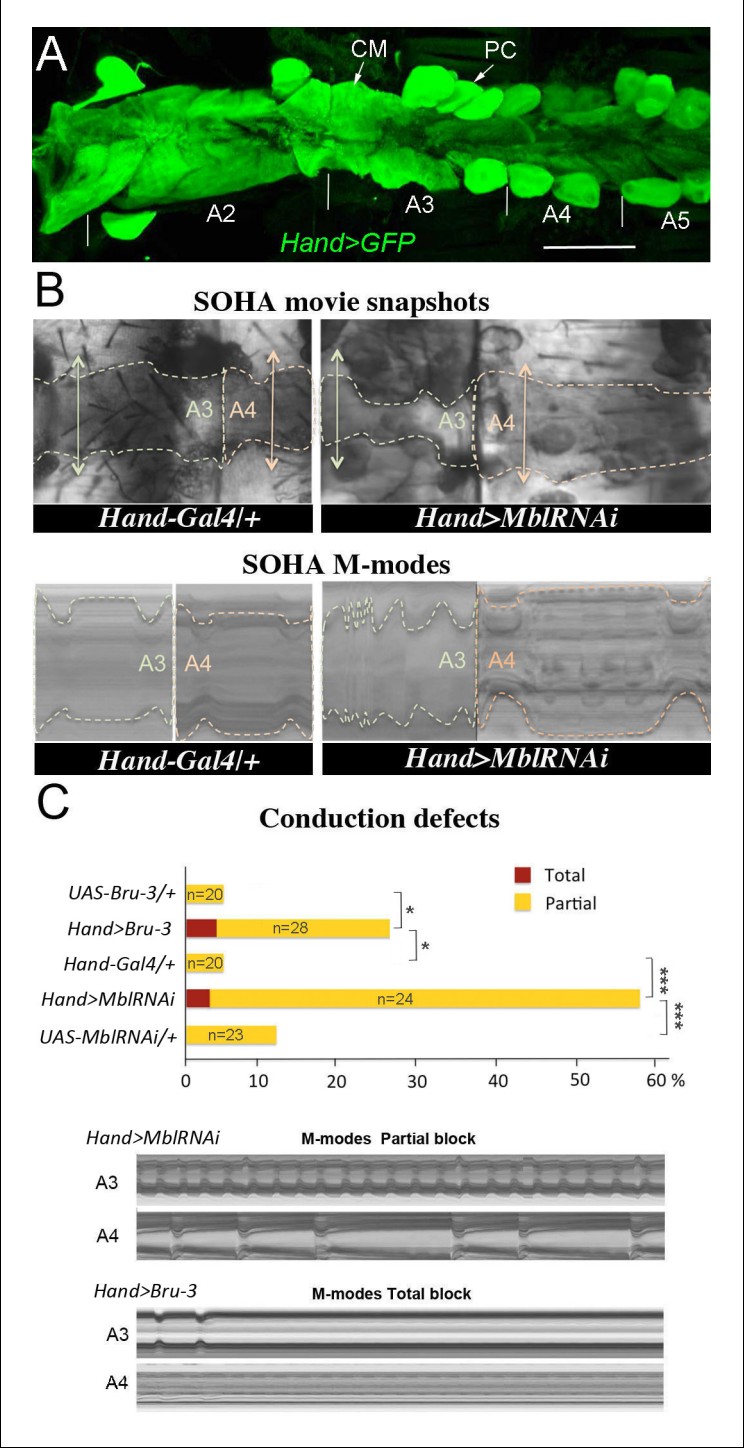

**Figure 1.** Cardiac-specific knockdown of MBNL1 ortholog and overexpression of CELF1 counterpart in *Drosophila* lead to asynchronous heartbeats. (**A**) The adult *Drosophila* heart expressing Hand-Gal4-driven GFP (*Hand>GFP*). Note that this Gal4 line drives expression exclusively in the heart. Arrows indicate cardiomyocytes (CM) and pericardial cells (PC) and A2–A5 denote abdominal segments. Scale bar, 150 μm. (**B**) Movie and M-modes views illustrating asynchronous heartbeats in *Hand>MblRNAi* flies registered in two adjacent heart segments (A3 and A4). Two-sided arrows indicate heart diameter in diastolic state. (**C**) Barplot graph showing percentage of flies with conduction defects in the different genetic contexts tested. Note the higher impact of attenuation of *Mbl* compared to overexpression of *Bru-3*. Number of fly hearts tested (n) is indicated and statistical significance

*Figure 1 continued on next page*

*Figure 1 continued*

(Fisher's exact test) denoted by * (p<0.05) and *** (p<0.001). Below are examples of M-modes illustrating 'partial' and 'total' heart blocks observed in *Hand>MblRNAi* and *Hand>Bru-3* flies developing conduction defects.
The online version of this article includes the following video and figure supplement(s) for figure 1:

**Figure supplement 1.** Bru-3 and Mbl are expressed in the adult fly heart and their transcript and protein levels are affected in *Hand>Bru-3* and *Hand>MblRNAi* flies.
**Figure supplement 2.** Effect of simultaneous cardiac attenuation of *Mbl* and overexpression of *Bru-3* on conduction defects.
**Figure 1—video 1.** An example of asynchronous heartbeat phenotype observed in *Hand>MblRNAi* context.
https://elifesciences.org/articles/51114#fig1video1

expressing cardiac cells (*Figure 1—figure supplement 1*). The *Hand-GAL4* driven overexpression of *Bru-3* and attenuation of *Mbl*, both appeared homogenous along the heart tube.

To assess the cardiac function in these two DM1 models we used the semi-automated optical heartbeat analysis (SOHA) method, which consists in filming the adult beating heart from hemi-dissected flies and allows analysis of the contractile and rhythmic heart parameters (*Fink et al., 2009*; *Ocorr et al., 2007*). Besides frequently observed arrhythmia, about 58% of *Hand>MblRNAi* and 28% of *Hand>Bru-3* flies displayed asynchronous heartbeat (*Figure 1B,C* and *Figure 1—video 1*), a hallmark of conduction disturbance (*Birse et al., 2010*). In some cases, asynchronous heartbeat can result in total heart block when at least one heart segment stops contracting. About 3% of *Hand>MblRNAi* and 4% of *Hand>Bru-3* flies did develop a total heart block (*Figure 1C*). We also tested heartbeat in flies in which *Mbl* was attenuated in parallel to *Bru-3* overexpression (*Figure 1—figure supplement 2*) and found a moderate additive effect on conduction defects of these two deregulations present in DM1. Taken together, these results show that the imbalance of Mbl/MBNL1 and Bru-3/CELF1 in the fly heart increases the occurrence of asynchronous heartbeat. This prompted us to perform transcriptional profiling of *Hand>MblRNAi* and *Hand>Bru-3* in cardiac cells to identify misregulated gene expression underlying this phenotype.

## Transcriptional profiling of cardiac DM1 *Drosophila* models using heart-specific TU-tagging approach

Transcriptional profiling of cardiac cells from *Hand>MblRNAi* and *Hand>Bru-3* flies was performed using a cell-specific approach called TU-tagging (*Miller et al., 2009*), followed by RNA-seq. This method allows tissue-specific RNA isolation by expressing the uracil phosphoribosyltransferase (UPRT) enzyme cell-specifically and incorporation into neo-synthesized transcripts of a modified uracil, 4TU. Here, *Hand-GAL4*-driven UPRT expression in cardiac cells enabled us to isolate transcripts from adult heart cells only and we combined *UAS-UPRT* with *UAS-MblRNAi* and *UAS-Bru-3* lines to perform TU-tagging in these two DM1 contexts (*Figure 2A*). Comparison of RNA-seq data from input RNA fraction and heart-specific TU-fraction from control *Hand>LacZ;UPRT* flies was used to assess the suitability of TU-tagging in low-abundance heart cells. We found that transcripts of cardiac-specific genes such as *Hand* and *Tin* were highly enriched in the cardiac-tagged TU fractions, whereas the transcripts of *Npc1b* and *Obp19b* genes expressed in midgut (*Voght et al., 2007*) and gustatory cells (*Galindo and Smith, 2001*), respectively, were depleted (*Figure 2B*). Thus the TU-tagging method proved well-suited to the transcriptional profiling of adult *Drosophila* cardiac cells, allowing the identification of both global gene deregulations (analyzed here) and isoform-specific differential gene expression (in a parallel study).

## TU-tagging-based transcriptional profiling of DM1 models with heart asynchrony reveals aberrant expression of genes regulating cellular calcium transient

To select candidates with a potential role in conduction disturbance, we focused on a pool of genes commonly deregulated in *Hand>MblRNAi* and in *Hand>Bru-3* contexts (*Figure 3A*, Venn diagram), comprising 118 genes, with 64 conserved in humans (*Figure 3A*, heatmap and *Figure 3—source data 1*). Among them, gene interaction network analysis (*Figure 3B*) identified four candidates involved in the regulation of cellular calcium level and known to be essential for proper heart

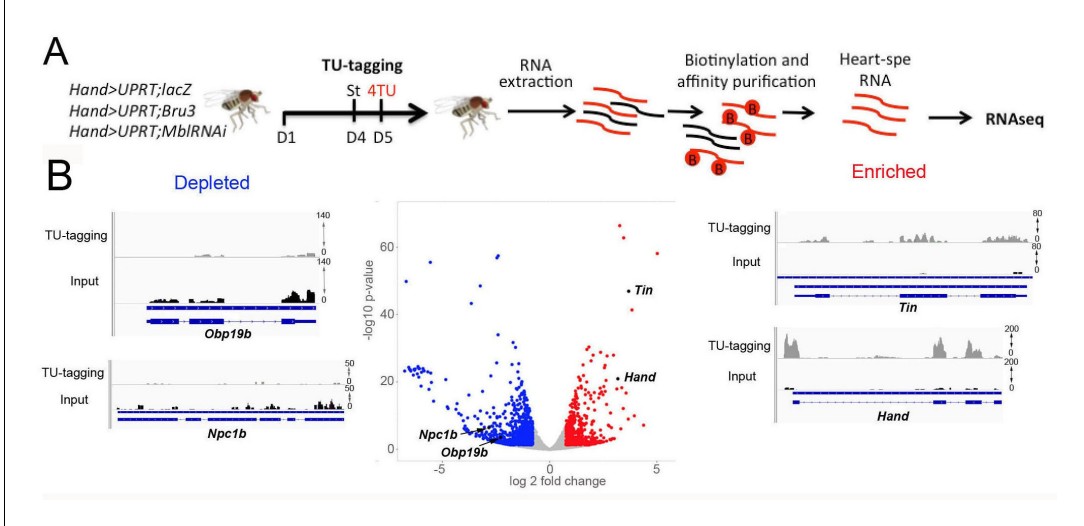

**Figure 2.** Cardiac-specific transcriptional profiling using TU-tagging method. (**A**) Pipeline of heart-targeted transcriptional profiling using TU-tagging. Note that Gal4- inducible UPRT transgene (*UAS-UPRT*) has been combined with *UAS-MblRNAi* and with *UAS-Bru-3* for the purpose of TU-tagging. *Hand>UPRT;lacZ* is the control line used to identify pathogenic gene deregulations in *Hand>UPRT;MblRNAi* and *Hand>UPRT;Bru-3* contexts. Flies were starved at day 4 for 6 hr before being transferred to 4TU containing food for 12 hr. (**B**) Volcano plot and IGV tracks from control *Hand>UPRT;lacZ* flies show examples of enrichment of heart-specific genes (e.g. *Hand*, *Tin*) (red, right side) and depletion of non-heart-expressed genes (blue, left side), thus validating the specificity of heart targeting.

function (**van Weerd and Christoffels, 2016**), namely *inactivation no afterpotential D* (*inaD*), *Syntrophin-like 1* (*Syn1*), *Rad, Gem/Kir family member 2* (*Rgk2*) and *straightjacket* (*stj*). Their respective human orthologs are *FRMPD2*, *SNTA1*, *REM1* and *CACNA2D3/α2δ3*. They all show high DIOPT orthology scores with their *Drosophila* counterparts (**Figure 3—source data 1**), except for *FRMPD2* encoding a scaffolding PDZ domain carrying protein structurally similar to inaD but with a low DIOPT score. As *inaD*, *Syn1* and *Rgk2* were downregulated in both DM1 contexts (**Figure 3A,B**) we assessed the effects of their attenuation. We observed that cardiac-specific knockdowns of *inaD*, *Syn1* and *Rgk2* overall led to an increase in diastolic and systolic diameters (**Figure 3—figure supplement 1**). However, we did not observe asynchronous heartbeat phenotypes, indicating that decrease in *inaD*, *Syn1* and *Rgk2* expression has no role in DM1-associated conduction defects. We also simulated increased cardiac expression of *stj* observed in both *Hand>MblRNAi* and in *Hand>Bru-3* contexts and found that in addition to arrhythmia and asystoles, *Hand>stj* flies exhibited asynchronous heartbeat, reminiscent of conduction defects observed in DM1 (**Figure 1C**). This observation and the fact that the expression of two other *Drosophila* α2δ genes (*CG42817* and *CG16868*) remained unchanged (**Figure 3—figure supplement 2**) prompted us to test whether elevated expression of *stj* could contribute to DM1 cardiac defects.

To link calcium transient to heart asynchrony, we first tested the propagation of calcium waves in normal and in asynchronously beating hearts. We applied the GCaMP3 fluorescent calcium sensor that allows detection of calcium waves in vivo (**Limpitikul et al., 2018**). As shown in **Figure 4A**, calcium waves perfectly shadow cardiac contractions in wild-type hearts, each calcium peak aligning with the beginning of contraction along the heart tube (here registered in segments A3 and A4). By contrast, in asynchronously beating *Hand>MblRNAi* hearts, calcium peaks are not always in perfect alignment with contractions in the anterior A3 segment and are not detectable in the A4 segment that does not beat (**Figure 4A**). Calcium waves thus correlate with cardiac contractions in normal fly hearts, and are affected in asynchronously beating hearts, linking calcium current regulation and DM1-associated conduction defects.

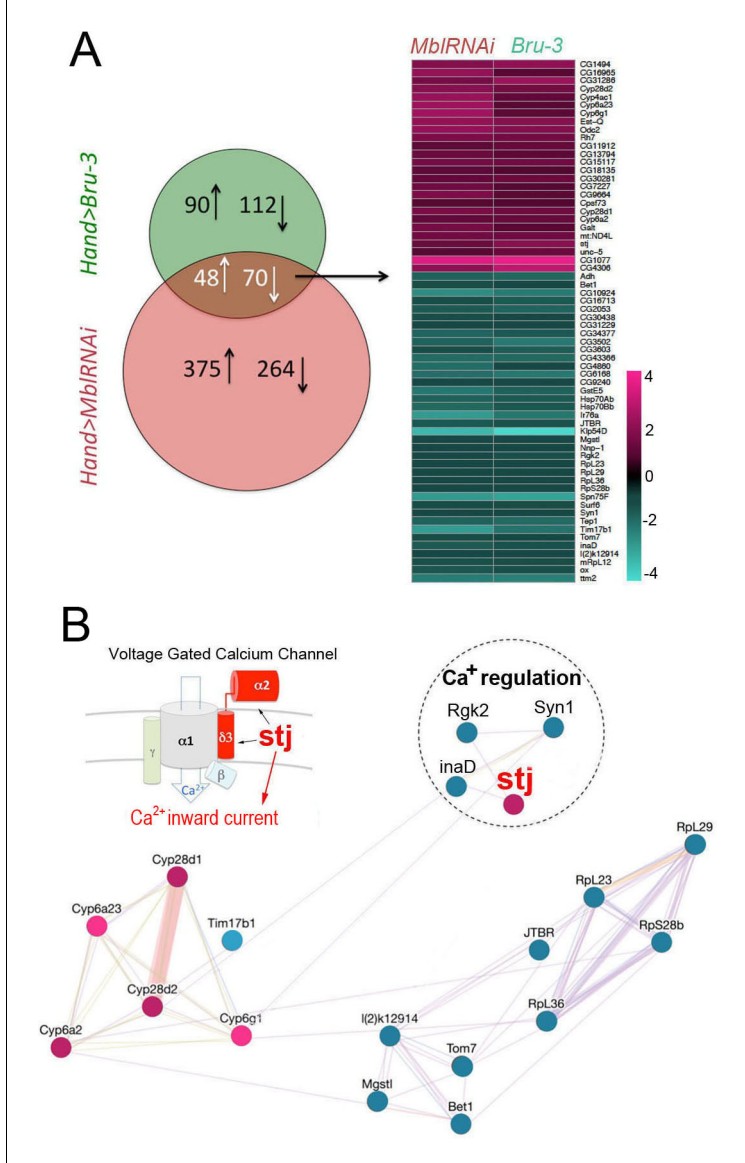

**Figure 3.** Heart-specific transcriptional profiling of DM1 flies identifies deregulation of genes controlling cellular calcium level. (A) Venn diagrams of genes deregulated in *Hand>UPRT;MblRNAi* and in *Hand>UPRT;Bru-3* contexts (FC > 1.7) followed by heatmap of commonly deregulated genes. (B) Genemania interaction network of conserved candidates including *stj*, *Rgk2*, *Syn1* and *inaD* known to be involved in $Ca^2$ regulation. A scheme presenting the structure of the voltage-gated calcium channel and its regulatory component Stj/α2δ3 is included. Color code in genemania network represents up and down regulation according to the heatmap. Cyan/Blue intensity code indicates fold change of genes expression in the heatmap.

The online version of this article includes the following source data and figure supplement(s) for figure 3:

**Source data 1.** A list of *Drosophila* genes and their Human orthologs deregulated in both *Hand>Bru-3* and *Hand>MblRNAi* contexts.

**Figure supplement 1.** Attenuation of *inaD*, *Syn1* and *Rgk2* all lead to an increase of diastolic (DD) and systolic (SD) heart diameters.

**Figure supplement 2.** The expression levels of *CG42617* and *CG16868*, two additional α2δ protein-coding genes are not affected in the heart of DM1 fly models.

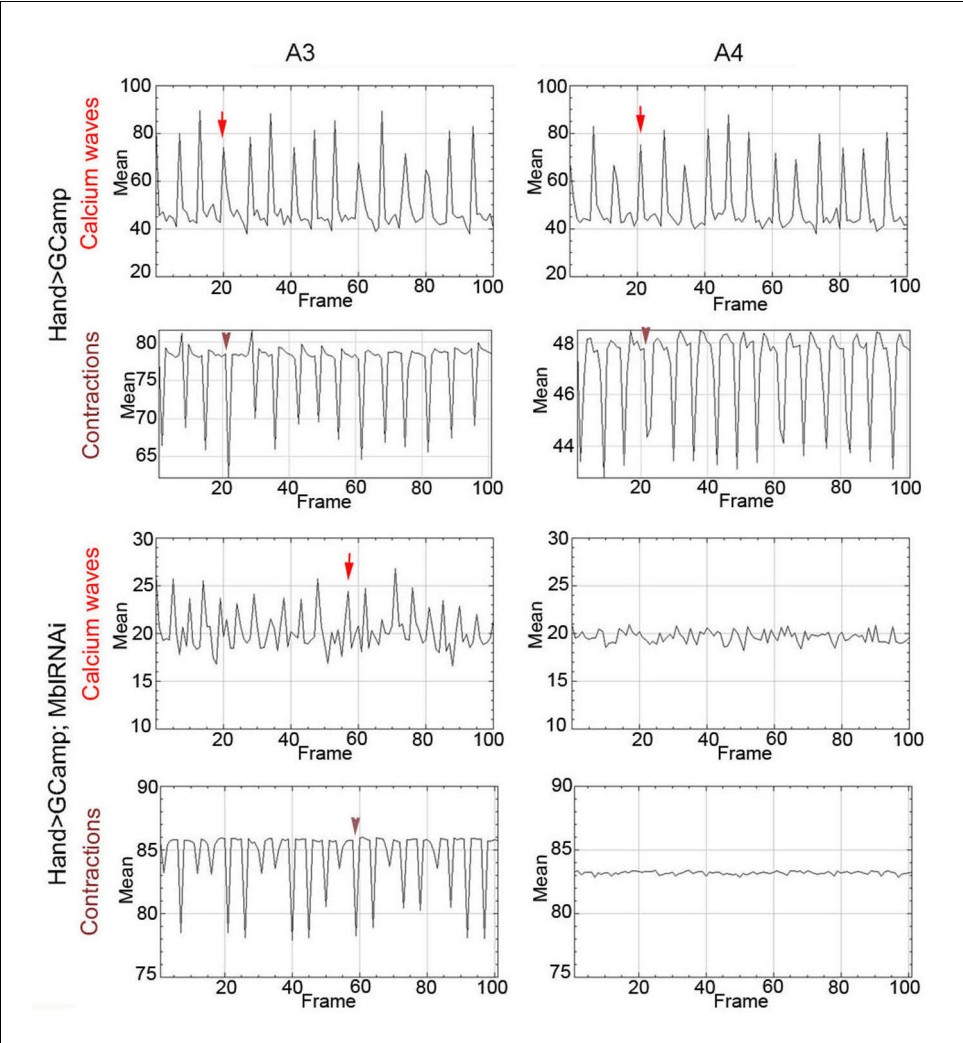

**Figure 4.** Calcium waves in asynchronous *Drosophila* heart. In wild type hearts, calcium waves underlie cardiac contractions so that calcium peaks (arrows) align with the onset of contractions (arrowheads) in both A3 and A4 segments. In contrast, in asynchronously beating *Hand>MblRNAi* heart, in the anterior A3 segment, calcium peaks (arrows) are not always in perfect alignment with contractions (arrowheads) and could not be detected in A4 segment that does not beat. Notice that GFP signal in *Hand>MblRNAi;GCaMP* context is lower than in the control (*Hand>GCaMP*) most probably because of a lower *Hand-Gal4* induction in two UAS transgene context.

## Increased cardiac expression of *Stj* contributes to asynchronous heartbeats in DM1 flies

To determine the cardiac function of Stj, we first tested whether this protein actually accumulated in the adult fly heart. A weak cytoplasmic Stj signal associated with circular myofibers of cardiomyocytes and a strong signal in the myofibers of ventral longitudinal muscle (VLM) underlying the heart tube (*Rotstein and Paululat, 2016*) was visible in the adult *Drosophila* heart (*Figure 5A* and scheme *Figure 5B*) stained with anti-Stj antibody (*Neely et al., 2010*). This suggested that low *stj* levels in cardiac cells could have functional significance. To test this hypothesis, we overexpressed *stj* in the entire adult fly heart using *Hand-Gal4* and in cardiomyocytes only using *Tin-Gal4* (*Figure 5C*). When driven with *Hand-Gal4*, about 25% of individuals overexpressing *stj* displayed conduction defects, and the percentage of flies with asynchronous heartbeats increased to 30% in the *Tin>stj* context (*Figure 5C*). This suggests that in flies, a high Stj level in cardiomyocytes creates a risk of conduction disturbance.

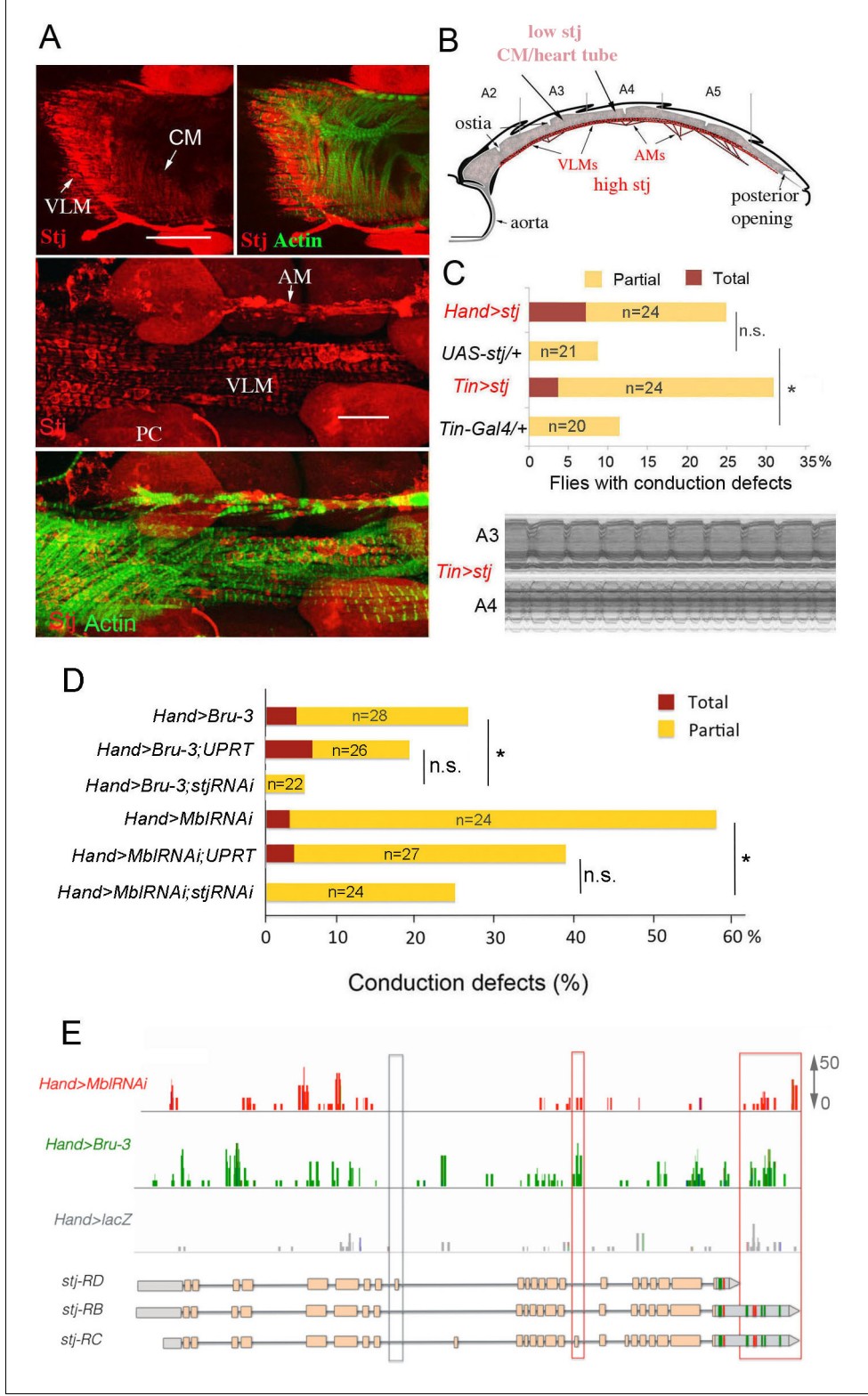

**Figure 5.** Increased cardiac expression of Stj contributes to the conduction defects observed in DM1 fly models. (**A**) Stj protein is detected at a low level in cardiomyocytes (CM) but at a higher level in the ventral longitudinal muscles (VLMs) that underlie adult heart tube. Stj in VLMs and in heart tube-attaching alary muscles (AMs) marks the network of T-tubules. PC denotes pericardial cells. Scale bars, 50 μm. (**B**) Scheme of the adult *Drosophila* heart with representation of Stj expression. (**C**) Both *Hand-Gal4* (whole heart) and *Tin-Gal4* (cardiomyocytes only) driven *Figure 5 continued on next page*

*Figure 5 continued*

overexpression of *Stj* lead to asynchronous heartbeats similar to those observed in *Hand>MblRNAi* and *Hand>Bru-3* contexts. (D) Barplot representing percentage of *Hand>MblRNAi* and *Hand>Bru-3* flies displaying asynchronous heartbeats after reducing cardiac *Stj* expression via RNAi (in *stj* rescue conditions). Notice that lowering Stj expression efficiently reduces the risk of asynchronous heartbeats in *Hand>Bru-3* context. Number of fly hearts tested (n) is indicated and statistical significance (Fisher's exact test) denoted by ns (p>0.05) and * (p<0.05). (E) IGV tracks showing RNAseq peaks over the *stj* locus in healthy control and pathological contexts. Red boxes highlight *stj-RC* – specific exon 15 tracks enriched in *Hand>MblRNAi* and *Hand>Bru-3* contexts and 3'UTR-specific tracks.

The online version of this article includes the following source data and figure supplement(s) for figure 5:

**Source data 1.** Corrected total cell fluorescence (CTCF) of Stj signal in circular fibers of cardiomyocytes.
**Figure supplement 1.** Stj protein levels in different genetic contexts visualized by immunostaining.
**Figure supplement 2.** *stj-RC* transcript isoform carrying long 3'UTR and alternatively spliced exon 15 is up regulated in *Hand>MblRNAi* and in *Hand>Bru-3* hearts.

To test whether *stj* is a required mediator of *Bru-3*-overexpression and/or *MblRNAi*-induced conduction defects, we performed genetic rescue experiments by attenuating *stj* expression *via* RNAi in *Hand>Bru-3* and *Hand>MblRNAi* flies (*Figure 5D*). These experiments revealed reductions of asynchronous heartbeats, which were statistically significant for both DM1 contexts but non-significant when referenced to DM1/UPRT lines, showing lower conduction defects (*Figure 5D*). Intriguingly, lowering *stj* transcript levels in the heart rescued asynchronous heartbeat more efficiently in *Hand>Bru-3* flies than in *Hand>MblRNAi* flies (*Figure 5D*). This suggests that fine-tuning of $Ca^2$ level by *stj*, but potentially also other calcium regulators, may be at work to ensure synchronous heartbeat and avoid conduction block.

We then set out to correlate Stj protein expression level in *Hand>Bru-3, Hand>MblRNAi* and *Hand>stj* fly hearts with the extent of cardiac asynchrony in these contexts. We found that Stj signal in the cardiomyocytes was higher in *Hand>stj* flies than in *Hand>MblRNAi* and *Hand>Bru-3* flies (*Figure 5—figure supplement 1*). However, the percentage of flies displaying conduction defects was higher in the *Hand>MblRNAi* context than in the *Hand>stj* context (*Figure 5C,D*) indicating that Stj was not the sole factor whose deregulation causes asynchronous heartbeat in the *Hand>MblRNAi* context. This observation is consistent with partial rescue of conduction defects in the *Hand>MblRNAi;stjRNAi* context (*Figure 5D*).

We also undertook to determine, which among three *stj* transcript isoforms (*Figure 5E*) is elevated in DM1 contexts. The analysis of RNAseq peaks over the *stj* locus (*Figure 5E*) indicated that transcripts harboring exon 15 (*stj-RC*) and long 3'UTR sequences (*stj-RB* and *stj-RC*) are enriched in both *Hand>Bru-3* and *Hand>MblRNAi* fly hearts. The upregulation of *stj-RC* was validated by RT-qPCR on dissected wild type and DM1 hearts (*Figure 5—figure supplement 2*) showing that its expression level is equivalent to the expression of all *stj* transcripts thus providing evidence that *stj-RC* is the main *stj* isoform whose expression increases in pathological contexts.

## Ventricular cardiomyocytes of DM1 patients with conduction disturbance show an increased expression of α2δ3

To determine whether our data from the fly model were relevant to DM1 patients, we first tested α2δ3 protein and transcript levels in ventricles and atria of normal mouse hearts. Like Stj, α2δ3 protein was also detected in the mouse cardiac cells, with high levels in the atrial and low levels in the ventricular cardiomyocytes (*Figure 6A*). This differential α2δ3 expression in atria and ventricles was also found at the transcript level (*Figure 6B*).

We then analyzed α2δ3 transcript and protein expression in a restricted number of human ventricular muscle samples from healthy donors and from DM1 patients with conduction defects. Ventricular α2δ3 transcript and protein levels were both significantly higher in DM1 patients with conduction disturbances than in controls (*Figure 6C,D*). Analyzing the same human samples, we also found a higher expression of the main calcium channel α1/Cav1.2 unit (*Figure 6—figure supplement 1*), further evidence (*Figure 6E*) that Ca2 inward current deregulation underlies DM1-associated conduction defects.

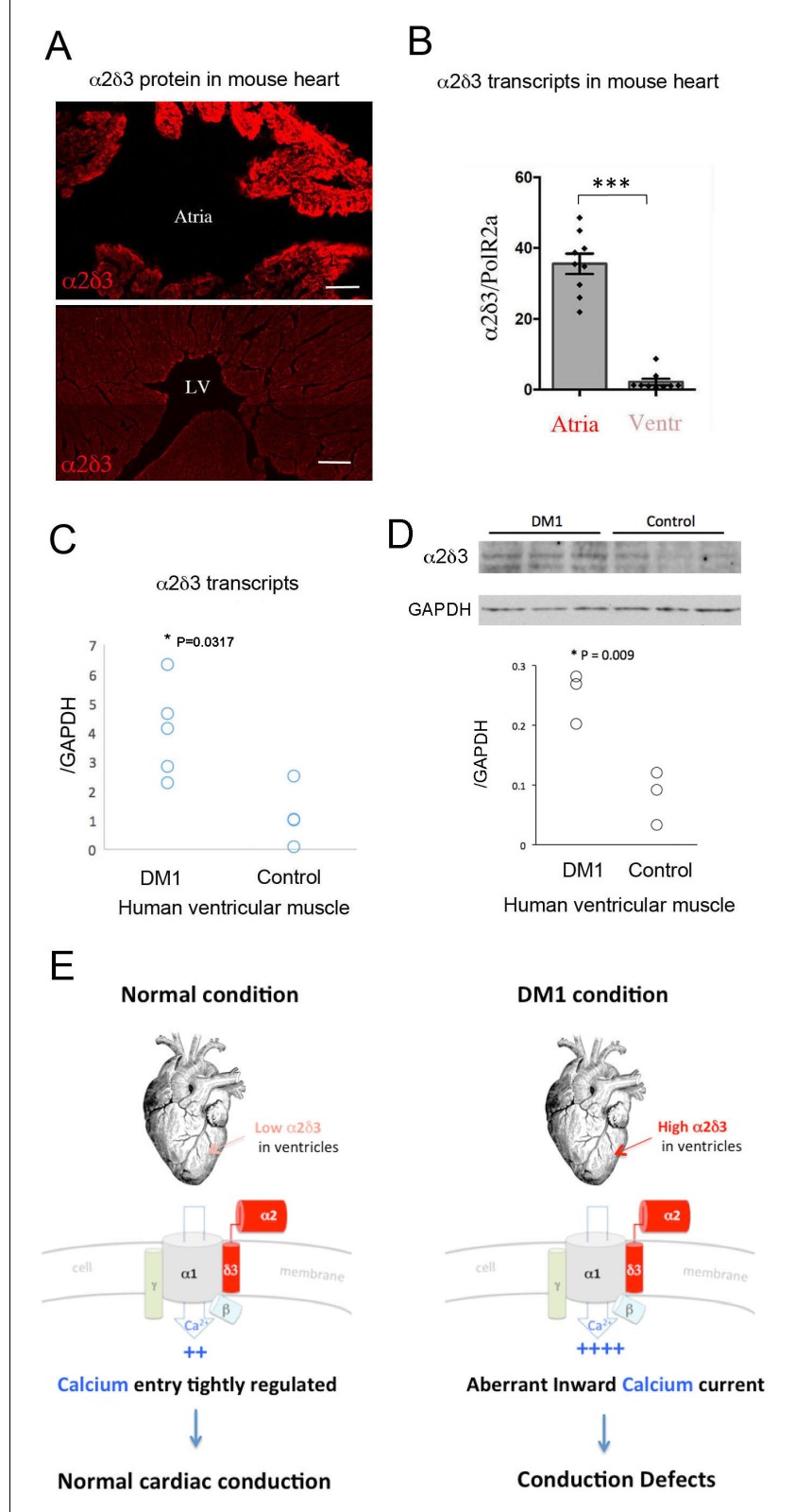

**Figure 6.** Increased cardiac expression of Straightjacket human ortholog $\alpha2\delta3$ is associated with conduction defects in DM1 patients. (**A**) In mouse, $\alpha2\delta3$ protein is detected at a high level in atrial cardiac cells and at a low level in ventricles. LV denotes left ventricle. Scale bars, 100 mm. (**B**) RT-qPCR analysis of mouse $\alpha2\delta3$ transcripts confirms low expression level in ventricles and high expression level in atria. Statistical significance was

*Figure 6 continued on next page*

*Figure 6 continued*

determined by Mann-Whitney U test (*** p<0.001) (**C, D**) α2δ3 (**C**) transcript and (**D**) protein levels analysed by RT-qPCR (**C**) and western blot (**D**) in human ventricular muscle from normal controls and DM1 patients with conduction defects. Note the statistically relevant increase in both α2δ3 transcript and α2δ3 protein levels in ventricles of DM1 patients (Mann-Whitney U test). (**E**) A scheme illustrating normal and DM1 conditions with low and increased α2δ3 levels in ventricular muscle, respectively. In pathological context, the aberrant inward calcium current in ventricular cardiomyocytes could lead to conduction defects and in particular to the IVCD.

The online version of this article includes the following figure supplement(s) for figure 6:

**Figure supplement 1.** RT-qPCR analysis showing that Cav1.2 transcripts are elevated in human cardiac samples from DM1 patients with conduction defects.

## Discussion

Cardiac dysfunctions decrease life expectancy in DM1, and conduction disturbances affect up to 75% of DM1 patients (*McNally and Sparano, 2011*). They are diagnosed by a lengthened ECG PR segment indicating atrio-ventricular (AV) block or QRS complex widening (*Groh et al., 2008*) that might or might not be associated with bundle-branch blocks of the His-Purkinje conduction system. Lengthened QRS without complete or incomplete bundle-branch block results from IVCD. It involves affected conduction via Purkinje fibers but also via ventricular cardiomyocytes that could contribute to the QRS widening observed in DM1 patients.

To identify gene deregulations underlying conduction defects in DM1 patients, we used the *Drosophila* model, which harbors a simple heart tube without a specialized conduction system, implying that synchronous propagation of contraction waves involves cardiomyocytes only. We reasoned that this model could provide insights into IVCD associated with ventricular cardiomyocyte dysfunction.

We applied a heart cell-specific transcriptional profiling approach based on TU-tagging (*Miller et al., 2009*) by which we identified 64 conserved genes commonly deregulated in both DM1 contexts with asynchronous heartbeat. Applying a sensitive cell-specific approach followed by RNA-seq was critical for revealing discrete gene deregulations, considering that cardiac asynchrony occurred only in a subset of DM1 flies. Also, *Hand>MblRNAi* and *Hand>Bru-3* flies developed two other DM1-associated heart phenotypes: dilated cardiomyopathy and arrhythmias. To facilitate candidate gene selection, we generated gene interaction networks for deregulated candidates and found that four of them were involved in the regulation of cellular calcium level.

### Calcium regulators and heart function

We previously reported (*Picchio et al., 2013*) that abnormal splicing of SERCA encoding the major SR calcium pump was involved in the myotonia phenotype in DM1 flies. Analyzing two heart-specific DM1 fly models, we found that four other conserved calcium regulatory genes, *inaD/FRMPD2*, *Syn1/SNTA1*, *Rgk2/REM1* and *stj/α2δ3*, are commonly deregulated. Genes *inaD* and *Syn*, which were downregulated, encode scaffolding PDZ domain proteins with regulatory functions on TRP $Ca^2$ channels (*Shieh and Zhu, 1996*; *Ueda et al., 2008*); their mutations cause cardiac hypertrophy, arrhythmia, and heart block (*Spassova et al., 2004*). Moreover, *inaD* was reported to prevent depletion of ER calcium store by inhibiting $Ca^2$ release-activated $Ca^2$ (CRAC) channels (*Su et al., 2003*). *Rgk2*, also downregulated, belongs to the Ras superfamily encoding small GTP-binding proteins (*Puhl et al., 2014*). Rgk proteins including Rgk2 interact with the α− or β-subunit of $Ca_V1.2$ calcium channel and negatively regulate its function (*Puhl et al., 2014*; *Magyar et al., 2012*). However, by interacting with CaMKII and with Rho kinase, both involved in cardiac hypertrophy and heart block, Rgk proteins could also influence heartbeat Cav1.2-independently. Finally, *stj*, whose transcript levels were significantly increased in both DM1 contexts, encodes an auxiliary subunit of this same $Ca_V1.2$ channel, and is known to positively regulate this channel's abundance (*Hoppa et al., 2012*). $Ca_V1.2$ mutations are known to lead to diverse heartbeat dysfunctions (*Splawski et al., 2004*), and $Ca_V1.2$ deregulation is associated with DM1 (*Rau et al., 2011*). Importantly, the *Drosophila* counterpart of Cav1.2 (Ca-1αD) also ensures cardiac contractions and calcium transients in the fly heart (*Limpitikul et al., 2018*), suggesting a conserved role of Cav1.2/Ca-1αD in heartbeat. Here, by following a fluorescent calcium sensor (GCaMP3), we found that calcium waves shadowed synchronous cardiac contractions but were disrupted in asynchronously beating DM1 fly hearts, implying that

aberrant calcium transient in cardiomyocytes is associated with heart asynchrony. Thus even if no specialized conduction system is present in the *Drosophila* heart, propagation of calcium waves along the cardiac tube is finely regulated to ensure synchronous heartbeat. Among selected calcium regulators, only cardiac overexpression of *stj* resulted in asynchronous heartbeat, so we focused our study on this candidate gene.

### *Stj/α2δ3*, new candidate gene for DM1-associated conduction defects

A previous large study dedicated to identifying genes involved in heat nociception (*Neely et al., 2010*) identified *stj/α2δ3* as an evolutionarily conserved pain gene. Here, we found *stj* transcripts upregulated in cardiomyocytes of two DM1 fly models *Hand>Bru-3* and *Hand>MblRNAi* exhibiting asynchronous heartbeat. We also show that reducing *stj* transcript levels significantly ameliorated heart asynchrony in *Hand>Bru-3* hearts while improving the conduction defect phenotype of *Hand>MblRNAi* flies. Thus additional gene deregulations contribute to asynchronous heartbeat in *Hand>MblRNAi* context, attesting the complexity of heart dysfunction in DM1. At the protein level, Stj is also more abundant in cardiomyocytes of two *Drosophila* DM1 contexts with cardiac asynchrony, while at a very low level in the wild-type cardiomyocytes. This suggests Stj protein involvement in the DM1-associated asynchronous heartbeat in flies. The precise mechanisms causing *stj* transcript elevation in DM1 contexts remain to be defined, but our data suggest that a combination of splice-dependent and 3'UTR-dependent mechanisms could be in play.

Intriguingly, vertebrate *stj* counterpart *α2δ3* also displays low transcript and protein levels in ventricular cardiomyocytes and a distinctively higher expression in atrial cardiac cells. We hypothesize that this differential expression could be related to the mechanisms regulating cardiomyocyte contraction in atria and in ventricles. The high *α2δ3* expression in atria is consistent with cardiomyocyte-dependent conduction in atria that do not harbor a specialized conduction system. By contrast, its low expression in ventricles could be correlated with the presence of His-Purkinje fibers facilitating conduction in the large ventricular muscles. We were, however, unable to test *α2δ3* involvement in cardiac conduction defects in a DMSXL mouse DM1 model carrying large numbers of repeats (*Huguet et al., 2012*) as it displays only mild cardiac involvement without conduction disturbances. We thus tested *α2δ3* expression directly in human cardiac samples. Ventricular cardiomyocytes from healthy donors showed, like the control mice, low *α2δ3* transcript expression, which was significantly elevated in ventricular cardiac cells from DM1 patients with conduction defects. DM1-associated increase of expression also occurs for *α2δ3* protein and is accompanied by high transcript levels of the main cardiac channel unit *Cav1.2/α1*, suggesting that the *α2δ3* elevation leads to a higher channel density and increased calcium entry to the cardiomyocytes. This could contribute to IVCD affecting the ventricular cardiomyocyte conduction rate and thereby the synchrony of cardiac contraction.

Taken together, with evidence based on heart-specific transcriptional profiling of DM1 *Drosophila* models and on gene deregulation in cardiac samples from DM1 patients, we propose that the conduction disturbance observed in DM1, and in particular IVCD, arises from likely impacted Ca-α1D/Cav1.2 calcium channel function in ventricular cardiomyocytes in which regulatory subunit stj/α2δ3 plays a central role.

## Materials and methods

### Key resources table

| | Reagent type (species) or resource | Designation | Source or reference | Identifiers | Additional information |
|---|---|---|---|---|---|
| 1 | | | | | |
| 2 | Gene (*D. melanogaster*) | *stj* | | FLYB: FBgn0261041 | |
| 3 | Gene (*D. melanogaster*) | *mbl* | | FLYB: FBgn0265487 | |
| 4 | Gene (*D. melanogaster*) | *bru3* | | FLYB: FBgn0264001 | |
| 5 | Genetic reagent (*D. melanogaster*) | *UAS-stj* | FlyORF Zurich ORFeome Project | FlyORF ID: F001252 | |

*Continued on next page*

*Continued*

| 1 | Reagent type (species) or resource | Designation | Source or reference | Identifiers | Additional information |
|---|---|---|---|---|---|
| 6 | Genetic reagent (*D. melanogaster*) | *stj TRIP* | Bloomington Drosophila Stock Center | RRID:BDSC25807 | RNAi line |
| 7 | Genetic reagent (*D. melanogaster*) | *Mbl RNAi* | VDRC Vienna Drosophila Resource Center | VDRC: GD 28732 | RNAi line |
| 8 | Genetic reagent (*D. melanogaster*) | *P{XP}bru3d09837* | Exelixis at Harvard Medical School | Harvard: d09837 | UAS carrying P element insertion line |
| 9 | Genetic reagent (*D. melanogaster*) | *UAS-GFP* | Bloomington Drosophila Stock Center | RRID:BDSC_32201 | |
| 10 | Genetic reagent (*D. melanogaster*) | *UAS-HA-UPRT* | Bloomington Drosophila Stock Center | RRID:BDSC_27604 | |
| 11 | Genetic reagent (*D. melanogaster*) | *UAS-lacZ* | Bloomington Drosophila Stock Center | RRID:BDSC_1776 | |
| 12 | Genetic reagent (*D. melanogaster*) | *UAS-GCaMP3* | Bloomington Drosophila Stock Center | RRID:BDSC_32116 | |
| 13 | Genetic reagent (*D. melanogaster*) | *Ca-alpha1D TRIP* | Bloomington Drosophila Stock Center | RRID:BDSC_25830 | RNAi line |
| 14 | Genetic reagent (*D. melanogaster*) | *Rgk2 TRIP* | Bloomington Drosophila Stock Center | RRID:BDSC_65932 | RNAi line |
| 15 | Genetic reagent (*D. melanogaster*) | *Rgk2 RNAi* | VDRC Vienna Drosophila Resource Center | VDRC: GD44611 | RNAi line |
| 16 | Genetic reagent (*D. melanogaster*) | *Syn1 RNAi* | VDRC Vienna Drosophila Resource Center | VDRC: GD27893 | RNAi line |
| 17 | Genetic reagent (*D. melanogaster*) | *Syn1 RNAi* | Bloomington Drosophila Stock Center | RRID: BDSC_27504 | RNAi line |
| 18 | Genetic reagent (*D. melanogaster*) | *inaD RNAi* | VDRC Vienna Drosophila Resource Center | VDRC: GD26211 (unavailable) | RNAi line |
| 19 | Genetic reagent (*D. melanogaster*) | *inaD RNAi* | VDRC Vienna Drosophila Resource Center | VDRC: 330227 | RNAi line |
| 20 | Genetic reagent (*D. melanogaster*) | *Hand-GAL4* | Laurent Perrin, TAGC, Marseille, France | | GAL4 driver line |
| 21 | Genetic reagent (*D. melanogaster*) | *TinCΔ4-GAL4* | Rolf Bodmer, SBPMD Institute, San Diego, US | | GAL4 driver line |
| 22 | Antibody | anti-Stj | Hugo Bellen, Baylor College of Medicine, Huston, USA | guinea pig polyclonal | Use 1:500 (IHC) |
| 23 | Antibody | anti-α2δ3 | Greg Neely, University of Sidney, Australia | rabbit polyclonal | Use 1:200 (IHC) |
| 24 | Antibody | anti-α2δ3 | GeneTex | GTX16618 - rabbit polyclonal | Use 1:500 (WB) |
| 25 | Antibody | anti-Bru3 | Millegen, Toulouse, France | rabbit polyclonal | Use 1:1000 (IHC) |
| 26 | Antibody | anti-Mbl | Darren Monckton, Glasgow, UK | sheep polyclonal | Use 1:200 (IHC) |
| 27 | Antibody | anti-GFP | Abcam | ab5450 – goat polyclonal | Use 1:500 (IHC) |
| 28 | Antibody | Anti-GAPDH | Sigma Aldrich | SAB2108266 – rabbit polyclonal | Use 1:3000 (WB) |

## *Drosophila* stocks

All fly stocks were maintained at 25°C on standard fly food in a 12 hr:12 hr light-dark cycling humidified incubator. To generate the DM1 *Drosophila* models, we used the inducible lines *UAS-MblRNAi*

(v28732, VDRC, Vienna, Austria) and *UAS-Bru3 37* (*bru-3^d09837*, BDSC, Bloomington, USA) and crossed them with the driver line, *Hand-Gal4* (*Han and Olson, 2005*; *Sellin et al., 2006*) (kindly provided by L. Perrin, TAGC, Marseille), to target transgene expression to the adult fly heart (cardiomyocytes and pericardial cells). To examine the expression pattern of the *Hand-Gal4* driver line, we crossed them to *UAS-GFP* line (#32201, BDSC). Control lines were generated by crossing the above-cited lines with *w^1118* flies. For TU-tagging experiments, the UPRT line (*UAS-HA-UPRT* 3.2, #27604, BDSC) was combined with the previously cited UAS lines and with *UAS-LacZ* (#1776, BDSC) as a control, generating *UAS-LacZ;UAS-HA-UPRT*, *UAS-MblRNAi;UAS-HA-UPRT* and *UAS-Bru3;UAS-HA-UPRT* stocks. For functional analyses of *straightjacket*, the lines used were *UAS-Stj* (FlyORF, F 001252) and *UAS-stj RNAi* (#25807, BDSC) and *tinCΔ4-Gal4* cardiomyocyte-specific driver (*Lo and Frasch, 2001*; *Perrin et al., 2004*) (kindly provided by R. Bodmer, SBPMD Institute, San Diego). For functional analyses of the candidate genes, we used the following UAS-RNAi lines: #26211 (VDRC) and #330227 (VDRC) for inaD; #27893 (VDRC) and #27504 (BDSC) for Syn1 and #44611 (VDRC) and #65932 (BDSC) for Rgk2. The UAS-GCaMP3 line (#32116, BDSC) was crossed to *Hand-Gal4* driver and applied as a sensor of calcium waves in the cardiac tube.

## Optical heartbeat analyses of adult *Drosophila* hearts

To assess cardiac physiology in adult flies, we used the method previously described (*Fink et al., 2009*; *Ocorr et al., 2007*). Briefly, 1 week old flies were anesthetized using FlyNap (# 173025, Carolina), and then dissected in artificial hemolymph solution, removing head, legs, wings, gut, ovaries and fat body. The hearts were allowed to equilibrate with oxygenation for 15–20 min prior to filming 30 s movies with a Hamamatsu camera (>100 frames/s) under a Zeiss microscope (10X magnification). Movie analysis was performed with SOHA (semi-automatic optical heartbeat analysis) based on using Matlab R2009b (Mathworks, Natick, MA, USA) to collect contractility (diastolic and systolic diameters, fractional shortening) and rhythmicity (heart period, diastolic and systolic intervals and arrhythmicity index) parameters.

## Immunofluorescence on *Drosophila* heart

Briefly, the fly hearts were dissected as described above and fixed for 15 min in 4% formaldehyde. The immunostaining procedure was performed as described elsewhere (*Picchio et al., 2013*). The following primary antibodies were used: sheep anti-Mbl antibody (1:200, kindly provided by Darren Monckton), rabbit anti-Bru-3 (1:1000; Millegen, Toulouse, France), rabbit anti-Stj (1:500; kindly provided by Hugo Bellen), rabbit anti-α2δ3 (1:200, gift of Greg Neely) and goat anti-GFP (1:500, Abcam, ab 5450). Rhodamine phalloidin (ThermoFischer Scientific) was used to reveal actin filaments. Fluorescent secondary antibodies were from Jackson ImmunoResearch. For Mbl, we used a biotinylated anti-sheep antibody (Biotin-SP-AffiniPure Donkey Anti-Sheep IgG (H L), Jackson ImmunoResearch) combined with a DTAF-conjugated streptavidin (Jackson ImmunoResearch).

## RNA extraction and RT-qPCR on adult fly heart samples

Total RNA was isolated from about 10–15 hearts from 1 week old flies, using TRIzol reagent (Invitrogen) combined with the Quick-RNA MicroPrep Kit (Zymo Research) following the manufacturer's instructions. RNA quality and quantity were respectively assessed using Agilent RNA 6000 Pico kit on an Agilent 2100 Bioanalyzer (Agilent Technologies) and Qubit RNA HS assay kit on a Qubit 3.0 Fluorometer (Thermo Fischer Scientific). RT-qPCR (three biological replicates) was performed as previously described (*Picchio et al., 2013*), using *Rp49* as a reference gene. The following pairs of primers were used: Rp49: forward GCTTCAAGGGACAGTATCTG and reverse AAACGCGGTTCTGCATGAG; Mbl: forward CGTGGAGGTCCAGAACGG and reverse AATATCAGGTCAATATAAAGCG; Bru-3: forward CCGAAGGTTGAGTTTGCTC and reverse CTTCAGCTGTAAAGCACGG; total *stj* transcripts: forward AGGCTCGCGAGTTCAACC and reverse AAGGCCTGCTCCGTGTAC and exon 15 carrying transcripts: forward GACGCCCTTTACTCTGGTCA and reverse ATAAAGGGCACAAAC TTGCG.

## Immunofluorescence on murine heart

Mouse hearts were dissected and fixed for 2 hr in paraformaldehyde (4% in 1X PBS) at 4°C and processed for cryosectioning as described elsewhere (*Beyer et al., 2011*). Frozen sections were

incubated overnight at 4°C with anti-α2δ3 (1:200) (*Neely et al., 2010*). Fluorescent images were obtained using a Zeiss AxioimagerZ1 with an Apotome module and an AxioCamMRm camera.

## RNA extraction and RT-qPCR on murine heart samples

RNA extraction was performed as described elsewhere (*Huguet et al., 2012*). Transcripts were amplified in a 7300 real-time PCR system (Applied Biosystems) using Power SybrGreen detection (Life Technologies). Standard curves were established with mouse tissues expressing the tested genes. Samples were quantified in triplicate, and experiments were repeated twice. α2δ3 transcripts were amplified using forward primer GGGAACCAGATGAGAATGGAGTC and reverse primer TTTGGAGAAGTCGCTGCCTG. *Polr2a* was used as internal control (forward primer GGCTG TGCGGAAGGCTCTG and reverse primer TGTCCTGGCGGTTGACCC).

## RNA extraction, RT-qPCR, and immunoblot on DM1 patient cardiomyocytes

Human ventricular cardiac muscle tissues were obtained at autopsy from five DM1 patients and four normal controls after informed consent was obtained. All experimental protocols were approved by the Institutional Review Board at Osaka University, and carried out in accordance with the approved guidelines. Total mRNA was extracted and first-strand complementary DNA synthesized using protocols described previously (*Nakamori et al., 2008*). RT-qPCR was performed using TaqMan Gene Expression assays (Hs01045030_m1 and 4333760F, Applied Biosystems) on an ABI PRISM 7900HT Sequence Detection System (Applied Biosystems), as described previously (*Nakamori et al., 2011*). Level of CACNA2D3/α2δ3 mRNA and Cav1.2/α1 mRNA were normalized to 18S rRNA. For protein analysis, cardiac muscle tissues were homogenized in a 10x volume of radioimmunoprecipitation assay buffer (25 mM Tris-HCl; pH 7.5; 150 mM NaCl; 1% NP-40; 1% sodium deoxycholate; and 0.1% sodium dodecyl sulfate) containing a protein inhibitor cocktail (Sigma-Aldrich). The homogenate was centrifuged for 10 min at 10,000 × g and the supernatant was collected. Equal amounts of protein (30 µg) were separated by sodium dodecyl sulfate polyacrylamide gel electrophoresis and transferred onto Immobilon-P membranes (Millipore), as previously described (*Nakamori et al., 2008*). Blots were blocked for nonspecific protein binding with 5% (w/v) nonfat milk and then incubated with a 1:500-diluted antibody against CACNA2D3/α2δ3 (Gene Tex) or 1:3000-diluted antibody against GAPDH (glyceraldehyde 3-phosphate dehydrogenase) (Sigma-Aldrich). After repeated washings, the membranes were incubated with horseradish peroxidase-conjugated goat anti-rabbit IgG (Life Technologies). The ECL Select Western Blotting Detection Reagent (GE Healthcare) and Chemi-Doc Touch Imaging System (Bio-Rad) were used to detect the proteins. Only the upper α2δ3 band (*Figure 6D*) corresponding to predicted protein size was quantified.

## TU-tagging experiments

The TU-tagging protocol was adapted from previously published studies (*Gay et al., 2014*; *Miller et al., 2009*).

### 4TU treatment, fly collection, and total RNA extraction

TU-tagging experiments were performed on the following 1-week-old adult flies: *Hand>LacZ;HA-UPRT*, *Hand>MblRNAi;HA-UPRT* and *Hand>Bru3;HA-UPRT*. Briefly, flies were starved for 6 hr at 25°C and transferred to fresh food media containing 4TU (Sigma) at 1 mM. After 12 hr incubation at 29°C, about 30–40 flies of the described genotypes were dissected (removing head, wings, legs, ovaries and gut) in DPBS 1X (Gibco, Life Technologies), immediately transferred to Eppendorf tubes, and snap-frozen in liquid nitrogen. Total RNA isolation was performed as described above in TRIzol following the manufacturer's instructions (ThermoFischer Scientific).

### Purification of TU-tagged RNA

Bio-thio coupling was performed on about 30 µg of total RNA with 1 mg/mL biotin-HPDP (Pierce) for 3 hr followed by two chloroform purification steps to eliminate the unbound biotin, as previously described (*Miller et al., 2009*). To verify the efficiency of the biotinylation step, an RNA dot blot was performed using streptavidin-HRP antibody. The streptavidin purification step then served to collect the thiolated-RNA fraction (cardiac-specific 4TU) and the unbound fraction (non-cardiac RNAs) using

a µMACS streptavidin kit (MACS, Miltenyi Biotec) following the manufacturer's instructions. The purified fraction was precipitated as previously described (*Miller et al., 2009*). RNA quality and quantity were assessed using Bioanalyzer and Qubit systems according to the manufacturer's instructions. RT-qPCR was performed to check for enrichment of cardiac-specific transcripts (*Hand* and *H15*) by comparing 4TU fraction relative to input fraction (i.e. the biotinylated fraction containing cardiac and non-cardiac RNAs), with *Rp49* gene used as reference.

## RNA sequencing

### Library preparation

Library preparation was performed on about 50 ng of cardiac-specific RNA from the previously described conditions, using the Ovation RNA-Seq Systems 1–16 for Model Organisms adapted to *Drosophila* (NuGEN, #0350–32) as per the manufacturer's instructions with a few modifications. First-strand synthesis was performed with the integrated DNAse treatment (HL-dsDNAse, Arctic-Zymes) and cDNA was fragmented using a Bioruptor sonication system (Diagenode) via 30 cycles of 30 s ON/OFF on low power. 15 PCR library amplification cycles were performed on 50 ng of starting RNA, and a size-selective bead purification step was done using RNAClean XP beads (Agencourt AMPure, Beckman Coulter). Quantitative and qualitative assessment of the library was performed using an Agilent dsDNA HS kit on an Agilent 2100 Bioanalyzer (Agilent Technologies).

### Sequencing parameters

The indexed Illumina libraries prepared from the cardiac-specific *Hand>LacZ;HA-UPRT*, *Hand>MblR-NAi;HA-UPRT* and *Hand>Bru3;HA-UPRT* RNA fractions were mixed at equal concentrations and sequenced (100 bp paired-end reads) on the same lane of a HiSeq 2000 system (EMBL Gene Core Illumina Sequencing facility, Heidelberg, Germany). All four genotype sets of 1-week-old flies were sequenced in duplicate to give a total of eight samples sequenced.

### Data analysis

The RNA-sequencing data were checked for good quality using the FastQC package (http://www.bioinformatics.babraham.ac.uk/projects/fastqc/). Each sample was aligned to the reference Dmel genome release 5.49 using Bowtie2 (*Langmead and Salzberg, 2012*). The aligned reads were sorted and indexed using SAMtools (*Li et al., 2009*). Pearson correlation was determined for each duplicate of each condition. Differential gene expression was obtained using a pipeline based on the Deseq2 package. Rlog transformation was applied on raw count data obtained for the duplicates of each condition before computing differential expression. Genes were tested individually for differences in expression between conditions. We set a fold-change threshold at 1.7 and *p*-value threshold at 0.05 for meaningful differential up-expression, and fold-change threshold at 0.59 and *p*-value threshold at 0.05 for meaningful differential down-expression.

## In vivo calcium transient analyses

We performed short (10 s) time lapse recording of beating hearts, dissected as for SOHA experiments, and mounted inverted on 35 mm glass-bottom dishes (IBIDI ref 81218) for imaging on LEICA SP8 confocal microscope. Twenty control hearts (Hand>GCaMP) and 35 DM1 hearts dissected from Hand>MblRNAi;GCaMP 1-week-old flies were analyzed. Two channel films recorded from a single optical level (GCaMP-GFP channel representing calcium waves and transmitted light channel representing contractions) were analyzed using Image J. For each film, we first transformed Frames on Image stacks and used 'Plot *Z* axis profile' function to generate graphs.

## Statistical analyses and access to RNAseq data

All statistical analyses were performed using GraphPad Prism v5.02 software (GraphPad Inc, USA). Fisher's exact test was applied to analyze statistical significance of conduction defect phenotypes, and Mann-Whitney *U* test to analyze RT-qPCR data from mouse and human heart samples and Western blot α2δ3 expression data in human samples. We also applied one-way ANOVA, Kruskal-Wallis Dunn's multiple comparison post-test to statistically analyze RT–qPCR data for *stj* isoforms and changes in diastolic and systolic heart diameters. All RNAseq data reported here were deposited with the GEO-NCBI tracking system under accession number #18843071.

## Acknowledgements

This work was supported by AFM-Téléthon (MyoNeurAlp Strategic Program), Agence Nationale de la Recherche (Tefor Infrastructure Grant), Fondation pour la Recherche Médicale (Equipe FRM Award) and the Intramural Research Grant for Neurological and Psychiatric Disorders (NCNP grant to MN).

## Additional information

### Funding

| Funder | Grant reference number | Author |
|---|---|---|
| AFM-Téléthon | MyoNeurAlp strategic program | Krzysztof Jagla |
| Agence Nationale de la Recherche | Tefor Infrastructure | Krzysztof Jagla |
| Fondation pour la Recherche Médicale | Equipe FRM | Krzysztof Jagla |
| National Center of Neurology and Psychiatry | Intramural Research Grant for Neurological and Psychiatric Disorders | Masayuki Nakamori |

The funders had no role in study design, data collection and interpretation, or the decision to submit the work for publication.

### Author contributions

Emilie Auxerre-Plantié, Yoan Renaud, Formal analysis, Investigation, Methodology, Writing—original draft; Masayuki Nakamori, Funding acquisition, Investigation, Methodology, Writing—original draft; Aline Huguet, Caroline Choquet, Teresa Jagla, Formal analysis, Investigation, Methodology; Cristiana Dondi, Methodology; Lucile Miquerol, Formal analysis, Methodology, Writing—review and editing; Masanori P Takahashi, Supervision, Funding acquisition; Geneviève Gourdon, Supervision, Investigation, Methodology, Writing—review and editing; Guillaume Junion, Supervision, Methodology, Writing—review and editing; Monika Zmojdzian, Conceptualization, Formal analysis, Investigation, Visualization, Methodology, Writing—review and editing; Krzysztof Jagla, Conceptualization, Formal analysis, Supervision, Funding acquisition, Investigation, Methodology, Writing—review and editing

### Author ORCIDs

Emilie Auxerre-Plantié https://orcid.org/0000-0002-8703-7636
Yoan Renaud http://orcid.org/0000-0002-4036-8315
Teresa Jagla https://orcid.org/0000-0002-9277-6089
Monika Zmojdzian https://orcid.org/0000-0001-6174-2719
Krzysztof Jagla https://orcid.org/0000-0003-4965-8818

### Ethics

Human subjects: Human ventricular cardiac muscle tissues were obtained at autopsy from five DM1 patients and four normal controls after informed consent was obtained. All experimental protocols were approved by the Institutional Review Board at Osaka University (Protocol Number 14183-6) and carried out in accordance with the approved guidelines.

### Decision letter and Author response

Decision letter https://doi.org/10.7554/eLife.51114.sa1
Author response https://doi.org/10.7554/eLife.51114.sa2

## Additional files

### Supplementary files
• Transparent reporting form

### Data availability
All data generated or analysed during this study are included in the manuscript and supporting files. Sequencing data have been deposited with the GEO-NCBI tracking system under accession number GSE109370.

The following dataset was generated:

| Author(s) | Year | Dataset title | Dataset URL | Database and Identifier |
|---|---|---|---|---|
| Plantie E, Picchio L, Renaud Y | 2018 | Deregulation associated with cardiac conduction defects in Myotonic Dystrophy Type 1 using TU-Tagging | https://www.ncbi.nlm.nih.gov/geo/query/acc.cgi?acc=GSE109370 | NCBI Gene Expression Omnibus, GSE109370 |

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
