## [Decision Letter]

**Acceptance summary:**

This paper provides new insights into the underlying causes of cardiac defects in myotonic dystrophy. The authors use the fruit fly *Drosophila* as a model system which permits genetic manipulation of the misbalance between two RNA binding proteins, which triggers the disease. They examined the perturbations in gene expression in the heart that this produced and identified genes controlling calcium levels and notably a subunit of a calcium channel, α2δ3 known in *Drosophila* as straightjacket, which is present at an abnormally high level. They go on to show that overexpression of this gene in normal flies leads to asynchronous beating of the heart, whereas its downregulation in fly models for myotonic dystrophy improves the performance of the heart. Observations on mice and humans indicate that α2δ3 is normally low in the cardiac ventricles but is elevated in patients with myotonic dystrophy. These findings therefore suggest that reducing the level of this calcium channel sub-unit may be of therapeutic benefit for the prevention of cardiac conduction defects in patients suffering from myotonic dystrophy.

**Decision letter after peer review:**

[Editors’ note: a previous version of this study was rejected after peer review, but the authors submitted for reconsideration. The first decision letter after peer review is shown below.]

Thank you for submitting your work entitled "*Straightjacket/α2δ3* deregulation is associated with cardiac conduction defects in Myotonic Dystrophy type 1" for consideration by *eLife*. Your article has been reviewed by a Senior Editor, a Reviewing Editor, and three reviewers. The following individuals involved in review of your submission have agreed to reveal their identity: Rolf Bodmer (Reviewer #1); David Brook (Reviewer #3).

Our decision has been reached after consultation between the reviewers. Based on these discussions and the individual reviews below, we regret to inform you that your work will not be considered for publication in *eLife* as it stands. However, if you can respond to the comments then we encourage a resubmission of the revised manuscript which will be sent to the same reviewers. They and the editor thought that responding to their comments would take more than the statutory revision period. They appreciated the fly work, with some points to be addressed, but were more critical of the data showing a link with DM. In addition to improving the results on patient material, the link would be reinforced by showing that in the fly model that you had developed, with the expanded repeat RNA, similar effects on *stj* are seen as with CELF1 and MBNL1 changes in expression.

Reviewer #1:

This manuscript by Plantié et al., studies a potential role of straightjacket/α2δ3 deregulation in cardiac conduction defects in Myotonic Dystrophy Type 1 (DM1, taking advantage of the fruit fly as a genetic model. DM1 is a muscular dystrophy disease caused by a CTG triple repeat in *dystrophia myotonica protein kinase (dmpk*) which causes imbalance between Muscleblind-like 1 (MBNL1 (*Mbl* D. mel)) and CUGBP and ELAV-like family member 1 (CELF1 (*Bru-3* D. mel)). Importantly, as observed in flies the authors confirm transcriptional upregulation of α2δ3 in cardiac ventricles of DM1 patients supporting their findings. In their approach they used two different DM1 fly models exhibiting conduction defects and performed heart-specific TU-tagging based transcriptional profiling experiment, where they identified 4 genes involved in calcium handling (among them straightjacket, *stj*). Converging evidence from two different DM1 fly models with conduction defects that reveal a common subset of differentially expressed genes, demonstrates scientific rigor and strengthens the conclusions, including the link between the conduction defect and regulation of calcium transients. Overall, the findings are of high interest and relevance for understanding the mechanisms of DM1. However, several concerns need to be addressed.

Essential revisions:

1) Given that both genes together found to exhibit altered expression in DM1 patients (Mbl/MBNL1 down and Bru-3/CELF1 up) and that the fly models exhibit phenotype only in 25% (*Bru-3* OE) or 58% (*Mbl* KD) of the flies, it would be of interest, whether there is an additive or synergistic effect on heart function by combining *Mbl* KD and *Bru-3* OE.

2) More information is needed on the phenotypes of the other 3 genes, as they are potentially also contributing to the DM1-like phenotypes. For example, use different drivers for *inaD*, since lethal with *Hand-Gal4*.

3) Statistical methods need to be explained better and appropriately applied to all quantitative measurements. For example, the categorical data (how many flies show defects out of how many) need to be statistically evaluated, such as by Chi-square test (e.g. Figure 1C, Figure 5C,D), or corrected for non-normal distribution in other situations. In Figure 5D, is *Hand>MblRNAi;stjRNAi* statistically different from *Hand>MblRNAi* alone?

Reviewer #2:

The authors produced MBNL/CELF imbalance in *Drosophila* heart by decreasing the fly homologs Mbl/Bru-3. This produced asynchronous heart beats. They used heart-targeted TU-tagging specifically in heart cells followed by RNA sequencing to identify genes with altered expression that correlated with aberrant heart rhythm. They focus on *straightjacket (stj)*/CACNA2D3 (α2δ3), a subunit of the Cav1.2 voltage-gated calcium channel. Overexpression of *stj* in flies results in asynchronous heartbeats in 25-30% of flies and loss of *stj* function weakly rescues Mbl and rescues more convincingly rescues Bru-3.

One issue is the difficulty in establishing the specificity of the observed changes upon loss of Mbl or gain of Bru-3 as these relate to changes observed in DM1. For example, in both Mbl loss and Bru-3 gain, three other genes were shown to be down regulated and subsequent down regulation of each gene caused abnormal heart beats. MBNL and CELF families are highly conserved with antagonistic activities even in planaria so altered express will cause many abnormalities most of which are probably not related to DM. In this regard the study lacks experiments in a DM1 model where changes in straightjacket levels are observed in response to expression of CUG repeat RNA in the heart; CUG repeat RNA is the direct cause of the cardiac arrhythmias in DM1.

Another issue is that the level of Stj does not necessarily correlate with the extent of abnormal heart beats in Mbl knock down hearts (Figure 5C). While this could indicate additional contributing factors as suggested by the authors, it could also reflect a lack of cause effect relationship between MBNL/CELF, *stj* and cardiac arrhythmia.

The results attempting to show regulation of Stj mRNA by binding of Bru-3 is not convincing. There are no controls presented (mutation of putative binding sites) and what is the basis for identifying the Bru-3 and Mbl binding sites? A demonstration that Bru-3 regulates Stj mRNA levels by binding to the indicated element in the 3' UTR will need a good deal more work to establish the proposed hypothesis.

The statement that the results presented in Figure 6 "indicating a link between DM1 associated conduction defects and cardiac α2δ3 transcript regulation" is overstated. Given the large number of gene expression changes in DM1 heart, these data is not sufficient to make that case.

The data presented as immunofluorescence is difficult to interpret.

Reviewer #3:

Myotonic dystrophy type 1 is caused by a CTG repeat expansion in the 3'UTR of the DMPK gene. The repeat is transcribed but not translated into protein. The repeat expansion transcripts remain trapped in the nucleus where they sequester muscleblind-like proteins and stimulate the stabilization of CELF1. MBNL1 and CELF1 are counterbalanced factors involved in RNA splicing and an imbalance in these factors leads to a perturbation of RNA splicing in multiple transcripts and this is a key factor in DM1 pathophysiology. Electrocardiographic abnormalities are common in DM1, though this is rarely complete heart block.

In the present paper Plantié et al. identify Straightjacket (*stj*) deregulation as a factor in DM1-associated cardiac conduction defects. The authors overexpressed Bruno-3 (Bru-3) the fly ortholog of CELF1 and knocked down Mbl Muscleblind ortholog of MBNL1. Figure 1 shows the level of increase and or knockdown. In the case of Bru-3 this appears to be very highly expressed, whereas in DM1 CELF1 shows a comparatively modest increase and is stabilized via phosphorylation. Recent work indicates that number of repeat expansion transcripts is relatively small, possibly in the 6-20 range, which suggests that the amount of MBML1 sequestered may be relatively low.

Using *Drosophila* genetics the authors were able to perform heart-specific transcriptional profiling, which led to the identification of a subset of genes, including *stj*. The authors provide supporting evidence on calcium waves and rescue experiments for the relevance of this gene to heart conduction. However, is there any evidence that *stj* is mis-spliced as this is thought to be the primary consequence of the MBNL1/CELF1 imbalance? I am not aware of compelling evidence to support upregulation of genes as a consequence of MBNL1/CELF1 imbalance. A paper from Ebralidze et al. (2004) proposed RNA leaching of transcription factors in DM1 causing reduced gene expression, but this was not linked to MBNL1/CELF1 imbalance. In subsection “Increased cardiac expression of *stj* involves 3’UTR regulation and contributes to asynchronous heartbeats in DM1 flies”, the authors suggest a possible mechanism involving *stj* isoforms with extended 3'UTRs but it is unclear how this is contributing to increased *stj* expression.

The final part of the paper provides the key link to DM1 as the authors show increased α2δ3 (*stj* ortholog) transcript expression in ventricular biopsies. It would be reasonable to expect Western blot data to demonstrate increased protein levels and possibly some evidence of effects in tissues from other regions of the heart.

Overall this is an interesting observation but a little more work is needed to strengthen two aspects: first the molecular mechanism underlying an increase in *stj* levels and second evidence that the α2δ3 protein is increased in DM1 heart.

[Editors’ note: what now follows is the decision letter after the authors submitted for further consideration.]

Thank you for submitting your article "Straightjacket/α2‎δ3 deregulation is associated with cardiac conduction defects in myotonic dystrophy type 1" for consideration by *eLife*. Your article has been reviewed by Utpal Banerjee as the Senior Editor, a Reviewing Editor, and three reviewers. The following individuals involved in review of your submission have agreed to reveal their identity: Rolf Bodmer (Reviewer #2); David Brook (Reviewer #3).

The reviewers have discussed the reviews with one another and the Reviewing Editor has drafted this decision to help you prepare a revised submission.

The reviewers are largely satisfied with the considerable additional work and modifications introduced in this re-submitted manuscript. However, before publication they require you to address the following points. There are still concerns of over-interpretation. In particular, while it is reasonable to propose that Mbl and Bru-3 bind and regulate the *stj* 3' UTR, the evidence is just not there. Also, the evidence that exon 15 is regulated by Mbl and Bru-3 is not convincing. The western blots showing the increase in α2δ3 protein are helpful and add support for the contention that the ortholog is affected in the disease. Overall the results of the effects of altered Mbl and Bru-3 are well done and are worth reporting with several modifications.

The authors need to indicate in the manuscript that the expression of CUG RNA repeats in the fly model does not impact *stj* expression. Also putting this information in the context that the DMSXL mouse model does not show conduction defects does not provide a justification for the lack of an effect on *stj* in their fly model; rather it means that the mouse model is not representative of DM1 heart phenotype.

The RNAseq tracts in Figure 5E are not sufficient to demonstrate a change in splicing of the exon indicated by the authors. The sequencing is not sufficiently deep to even detect all of the constitutive exons in their entirety. This analysis requires RT-PCR across the alternative exon. The RT-PCR in Figure 5E—figure supplement 2 is not the standard (or correct) approach to determine PSI – this requires using primers on flanking constitutive exons. The authors claim that Bru-3 gain or Mbl protein loss 'promotes' inclusion of exon 15 specific for *stj-RC*. However, since "*stj* total" is augmented by the same relative level, it is overall expression that is augmented, and in this approach does not speak to an alteration in splicing pattern. Exon 15 does not seem to be preferentially selected for, there is just overall more expression. The conclusions should reflect that. Alternatively, since the alternative splicing of *stj* is not a main point for the paper this part can be removed.

Figure 5E—figure supplement 2 is not sufficient to implicate that there are Mbl and Bru-3 binding sites in the indicated regions. There needs to be an analysis of statistical enrichment; one can find short sequences matching RNA binding protein binding motifs given they are short and variable. There is no evidence to support the contention that Mbl and/or Bru-3 regulate *stj* levels. This aspect too could be removed.

The request for more thorough statistics has been done in most instances, except for Figure 1—figure supplement 1A,B, Figure 5—figure supplement 1F and Figure 6B. In Figure 1C, brackets should indicate exactly which bar is compared which bar, not as a vertical line across three bars, which is confusing.

---

## [Author Response]

[Editors’ note: the author responses to the first round of peer review follow.]

Reviewer #1:[…]Essential revisions:1) Given that both genes together found to exhibit altered expression in DM1 patients (Mbl/MBNL1 down and Bru-3/CELF1 up) and that the fly models exhibit phenotype only in 25% (Bru-3 OE) or 58% (Mbl KD) of the flies, it would be of interest, whether there is an additive or synergistic effect on heart function by combining Mbl KD and Bru-3 OE.

As suggested we generated *UAS-Bru-3;UAS-MblRNAi* double transgenic line and scored asynchronous heartbeat phenotypes in *Hand>Bru-3;MblRNAi* flies. To take into account double UAS context we also scored conduction defect phenotypes in *Hand>Bru-33;UPRT* and *Hand>MblRNAi;UPRT* flies. We found an increase in percentage of flies showing asynchronous heartbeats (partial block) from 33.3% in *Hand>MblRNAi;UPRT* up to 38.5% in *Hand>Bru-3;Mbl1RNAi* flies. Thus, this analysis reveals a moderate additive effect of Bru-3 overexpression and Mbl knockdown on conduction defects phenotype (new Figure 1—figure supplement 2). We add an appropriate comment stating this in the Results section.

2) More information is needed on the phenotypes of the other 3 genes, as they are potentially also contributing to the DM1-like phenotypes. For example, use different drivers for inaD, since lethal with Hand-Gal4.

We are grateful for this comment as it allowed us to more accurately characterize *inaD, Syn1 and Rgk2* cardiac phenotypes. We tested additional RNAi lines for each of them (listed in Materials and methods section). Notice that for *inaD* we tested two additional RNAi lines and found they do not display lethality when crossed with Hand-Gal4. We thus took into account only the newly tested *inaD* lines and consider that the previously tested TRIP 52313 line generated an off target effect. Also notice that increased arrhythmia index and asystole initially reported for *Rgk2* and fibrillations for *Syn1* have not been confirmed by analyzing additional RNAi lines. In new Figure 3—figure supplement 2 we present observed phenotypes detected in at least two RNAi lines per gene. We report that cardiac-specific knockdowns of *inaD*, *Syn1* and *Rgk2* all lead to an increase in diastolic and systolic diameters a phenotype also observed in our fly DM1 models.

However, we do not observe asynchronous heartbeat phenotypes in *inaD*, *Syn1* and *Rgk2* RNAi *contexts* indicating that decrease in their expression has no role in DM1-associated conduction defects.

3) Statistical methods need to be explained better and appropriately applied to all quantitative measurements. For example, the categorical data (how many flies show defects out of how many) need to be statistically evaluated, such as by Chi-square test (e.g. Figure 1C, Figure 5C,D), or corrected for non-normal distribution in other situations. In Figure 5D, is Hand>MblRNAi;stjRNAi statistically different from Hand>MblRNAi alone?

As suggested by the reviewer we provide now statistical evaluation of asynchronous heartbeat data in all presented genetic contexts. We applied Fisher’s exact test. In Figure 5D*Hand>MblRNAi;stjRNAi* rescue was compared with *Hand>MblRNAi;UPRT* double UAS condition and it is not significantly different. However, it is (p=0,02), when compared with *Hand>MblRNAi* alone (as suggested by the reviewer). This is also true for *Hand-Bru-3;stjRNAi* rescue condition. We thus modified Figure 5D by adding comparisons of Stj rescues with *Hand>MblRNAi* alone and *Hand>Bru-3* alone.

We also applied one-way ANOVA, Kruskal-Wallis Dunn’s multiple comparison post-test to evaluate statistical significance of changes in diastolic and systolic heart diameters presented in new Figure 3—figure supplement 2.

Reviewer #2:[…]One issue is the difficulty in establishing the specificity of the observed changes upon loss of Mbl or gain of Bru-3 as these relate to changes observed in DM1. For example, in both Mbl loss and Bru-3 gain, three other genes were shown to be down regulated and subsequent down regulation of each gene caused abnormal heart beats.

Regarding relation to DM1, knockdown of Mbl mimics sequestration of Mbl while gain of Bru-3 mimics an increase of Bru-3 level, both observed in DM1. In our previous study (Picchio et al., 2018) by using transcriptional profiling we already demonstrated that 85% of genes deregulated in *Drosophila* DM1 model expressing 960 CTG repeats are common with Mbl RNAi and about 50% with Bru-3 gain of function context. We thus consider that analysing Mbl loss and Bru-3 gain of function and in particular genes commonly deregulated in these two contexts could provide valuable cues to DM1. This strategy was applied in the present study.

To better characterize cardiac phenotypes of three other candidate genes (*inaD, Syn1* and *Rgk2)* and avoid potential off target effects we analyzed heart physiology in additional RNAi lines for those genes. Newly generated data are included to revised Figure 3—figure supplement 2.

MBNL and CELF families are highly conserved with antagonistic activities even in planaria so altered express will cause many abnormalities most of which are probably not related to DM. In this regard the study lacks experiments in a DM1 model where changes in straightjacket levels are observed in response to expression of CUG repeat RNA in the heart; CUG repeat RNA is the direct cause of the cardiac arrhythmias in DM1.

This issue needs some clarifications indeed as not all cardiac phenotypes observed in DM1 patients are easily reproduced in animal models expressing toxic CTG repeats. This is the case of DM1 associated conduction defects which were previously described i double KO mice for MBNL1 and MBNL2 (Lee et al., 2013) and in cardiac CELF1 gain of function context (Koshelev et al., 2010) but not in DMSLX mice harboring long CTG repeats (Algalarrondo et al., 2015).

Like in mice, in *Drosophila*, cardiac conduction defects are only present in loss of Mbl and gain of Bru-3 contexts but not in CTG repeats DM1 model. This indicates that loss of MBNL1/Mbl and gain of CELF1/Bru-3 models are more adapted than CTG repeat DM1 models to reproduce this severe cardiac phenotype observed in DM1 patients. Taking this into account we directly asked whether the increase in *stj* expression (contributing to conduction defects in fly models) could be detected in cardiac samples of DM1 patients with conduction defects.

It turned out that this is the case both at Stj/CACNA2D3 transcript and as we show now by Western blot at protein levels as well. Consistent with lack of cardiac conduction phenotypes, we do not detect changes in *stj* expression in *Drosophila* CTG repeat DM1 model.

Another issue is that the level of Stj does not necessarily correlate with the extent of abnormal heart beats in Mbl knock down hearts (Figure 5C). While this could indicate additional contributing factors as suggested by the authors, it could also reflect a lack of cause effect relationship between MBNL/CELF, stj and cardiac arrhythmia.

As stated above, conduction defects have been previously observed in MBNL1/2 DKO (Lee et al., 2013) and CELF1 gain of function (Koshelev et al., 2010) mouse models and we show here that this DM1-associated cardiac phenotype is also present in analogous contexts in *Drosophila*. As suggested by partial rescue of Mbl knockdown Stj is not the sole player, but it appears to be an important contributor.

The results attempting to show regulation of Stj mRNA by binding of Bru-3 is not convincing. There are no controls presented (mutation of putative binding sites) and what is the basis for identifying the Bru-3 and Mbl binding sites? A demonstration that Bru-3 regulates Stj mRNA levels by binding to the indicated element in the 3' UTR will need a good deal more work to establish the proposed hypothesis.

We agree that presented data do not sufficiently support 3’UTR-based regulation of Stj expression and that amount of work required to establish our hypothesis is not compatible with the time frame of all other revisions. However, to make a step forward we performed in silico search for MBNL1 and CELF1 binding sites (Lambert et al., 2014) within the 3’UTR *stj* region. We found that the long 3’UTR sequence contains several predicted CELF1/Bru-3 and MBNL1/Mbl binding sites (Figure 5—figure supplement 2), suggesting that an interplay between Bru-3 and Mbl proteins bound to 3’UTR could regulate *stj* isoforms abundance. In parallel, as demonstrated by Exon15 targeted qRT-PCR analysis (Figure 5—figure supplement 2) the Mbl loss and Bru-3 gain of function, both promote inclusion of Exon 15 leading to the upregulation of *stj-RC*. Thus, we hypothesize that in DM1 context a combination of splice-dependent and 3’UTR-dependent mechanisms control elevation of *stj* transcripts in cardiac cells. We include an appropriate comment to the text.

The statement that the results presented in Figure 6 "indicating a link between DM1 associated conduction defects and cardiac α2δ3 transcript regulation" is overstated. Given the large number of gene expression changes in DM1 heart, these data is not sufficient to make that case.

We agree and modify this Results section as follow:

“Ventricular α2δ3 transcript and protein levels were both significantly higher in DM1 patients with conduction disturbances compared to controls (Figure 6C, D). We also found, by analyzing the same human samples a higher expression of the main Calcium channel α1/Cav1.2 unit (Figure 6—figure supplement 1) further supporting the view (Figure 6E) that Ca^2^ inward current deregulation underlies DM1-associated conduction defects.”

Reviewer #3:[…]In the present paper Plantié et al. identify Straightjacket (stj) deregulation as a factor in DM1-associated cardiac conduction defects. The authors overexpressed Bruno-3 (Bru-3) the fly ortholog of CELF1 and knocked down Mbl Muscleblind ortholog of MBNL1. Figure 1 shows the level of increase and or knockdown. In the case of Bru-3 this appears to be very highly expressed, whereas in DM1 CELF1 shows a comparatively modest increase and is stabilized via phosphorylation. Recent work indicates that number of repeat expansion transcripts is relatively small, possibly in the 6-20 range, which suggests that the amount of MBML1 sequestered may be relatively low.

We agree that knocking down Mbl or overexpressing Bru-3 will not perfectly reproduce DM1 condition and in consequence all aspects of gene deregulations observed in DM1. Nevertheless, loss of Mbnl1 and gain of CELF1 functions have already been used to model DM1 by other labs. In our previous study (Picchio et al., 2018) by using transcriptional profiling we demonstrated that 85% of genes deregulated in *Drosophila* DM1 model with muscle targeted expression of 960 CTG repeats are common with Mbl RNAi and about 50% with Bru-3 gain of function context. We thus consider that analysing Mbl loss and Bru-3 gain of function and in particular genes commonly deregulated in these two contexts could provide valuable cues to DM1.

Using Drosophila genetics the authors were able to perform heart-specific transcriptional profiling, which led to the identification of a subset of genes, including stj. The authors provide supporting evidence on calcium waves and rescue experiments for the relevance of this gene to heart conduction. However, is there any evidence that stj is mis-spliced as this is thought to be the primary consequence of the MBNL1/CELF1 imbalance? I am not aware of compelling evidence to support upregulation of genes as a consequence of MBNL1/CELF1 imbalance. A paper from Ebralidze et al. (2004) proposed RNA leaching of transcription factors in DM1 causing reduced gene expression, but this was not linked to MBNL1/CELF1 imbalance. In subsection “Increased cardiac expression of stj involves 3’UTR regulation and contributes to asynchronous heartbeats in DM1 flies”, the authors suggest a possible mechanism involving stj isoforms with extended 3'UTRs but it is unclear how this is contributing to increased stj expression.

We provide now a more compelling evidence for the up regulation of one particular stj isoform – *stj-RC* in both *MblRNAi* and Bru-3 gain of function contexts (Figure 5—figure supplement 2). Stj-RC carries long 3’UTR but also contains alternatively spliced Exon 15. As demonstrated by Wang et al., (2015), both MBNL1 and CELF1 do not only act as alternative splice regulators but also display 3’UTR binding activities with potential role in regulation transcripts stability. In the case of *stj-RC* transcript its 3’UTR sequence contains several predicted CELF1/Bru-3 and MBNL1/Mbl binding sites, suggesting that an interplay between Bru-3 and Mbl proteins bound to 3’UTR could regulate *stj-RC* abundance (Figure 5—figure supplement 2). In parallel, Bru-3 gain and Mbl loss, both promote inclusion of Exon 15 (Figure 5—figure supplement 2). Thus, we hypothesize that a combination of splice-dependent and 3’UTRdependent mechanisms is involved in up regulation of *stj*. We add appropriate comments to the text.

The final part of the paper provides the key link to DM1 as the authors show increased α2δ3 (stj ortholog) transcript expression in ventricular biopsies. It would be reasonable to expect Western blot data to demonstrate increased protein levels and possibly some evidence of effects in tissues from other regions of the heart.

We thank the reviewer for this suggestion. With limited amount of available tissue from cardiac biopsies we were able to perform Western blot on 3 out of 5 DM1 ventricular cardiac samples analysed for transcript expression. We found that similarly to transcripts, α2δ3 protein is significantly elevated in DM1 condition. These new data are included to Figure 6.

Overall this is an interesting observation but a little more work is needed to strengthen two aspects: first the molecular mechanism underlying an increase in stj levels and second evidence that the α2δ3 protein is increased in DM1 heart.

We thank the reviewer for this comment and hope our additional data included to Figure 5 and Figure 6 strengthen two above aspects.

[Editors' note: the author responses to the re-review follow.]

The authors need to indicate in the manuscript that the expression of CUG RNA repeats in the fly model does not impact stj expression.

We introduced the following sentence to the Introduction: “We did not observe asynchronous heartbeats in flies expressing in the heart 960CTG repeats. This DM1 model (Picchio et al., 2013) developed other cardiac phenotypes such as arrhythmia.”

Also putting this information in the context that the DMSXL mouse model does not show conduction defects does not provide a justification for the lack of an effect on stj in their fly model; rather it means that the mouse model is not representative of DM1 heart phenotype.

We agree. We are not making parallels between mouse DMSXL and our *Drosophila* DM1 – 960CTG model.

The RNAseq tracts in Figure 5E are not sufficient to demonstrate a change in splicing of the exon indicated by the authors. The sequencing is not sufficiently deep to even detect all of the constitutive exons in their entirety. This analysis requires RT-PCR across the alternative exon. The RT-PCR in Figure 5E—figure supplement 2 is not the standard (or correct) approach to determine PSI – this requires using primers on flanking constitutive exons. The authors claim that Bru-3 gain or Mbl protein loss 'promotes' inclusion of exon 15 specific for stj-RC. However, since "stj total" is augmented by the same relative level, it is overall expression that is augmented, and in this approach does not speak to an alteration in splicing pattern. Exon 15 does not seem to be preferentially selected for, there is just overall more expression. The conclusions should reflect that. Alternatively, since the alternative splicing of stj is not a main point for the paper this part can be removed.

We agree that alternative splicing is not a main point here and modify a portion of the text subsection “Increased cardiac expression of *stj* contributes to asynchronous heartbeats in DM1 flies” related to Figure 5E and Figure 5—figure supplement 2, which were also modified accordingly. Revised text reads as follow:

“We also undertook to determine, which among three *stj* transcript isoforms (Figure 5E) is elevated in DM1 contexts. The analysis of RNAseq peaks over the *stj* locus (Figure 5E) indicated that transcripts harboring exon 15 (*stj‐RC*) and long 3’UTR sequences (*stj-RB* and *stj‐RC*) are enriched in both *Hand>Bru‐3* and *Hand>MblRNAi* fly hearts. The up regulation of *stj‐RC* was validated by RTqPCR on dissected wild type and DM1 hearts (Figure 5—figure supplement 2) showing that its expression level is equivalent to the expression of all *stj* transcripts, thus providing evidence that *stj‐RC* is the main *stj* isoform whose expression increases in pathological contexts."

Figure 5E—figure supplement 2 is not sufficient to implicate that there are Mbl and Bru-3 binding sites in the indicated regions. There needs to be an analysis of statistical enrichment; one can find short sequences matching RNA binding protein binding motifs given they are short and variable. There is no evidence to support the contention that Mbl and/or Bru-3 regulate stj levels. This aspect too could be removed.

Having no evidence that some of predicted Mbl and/or Bru‐3 binding motifs present within 3’UTR *stj* region are functionally relevant we follow the reviewer’s suggestion and remove the map of potential Mbl and Bru‐3 binding motifs from Figure 5E and Figure 5—figure supplement 2A.

The request for more thorough statistics has been done in most instances, except for Figure 1—figure supplement 1A,B, Figure 5—figure supplement 1F and Figure 6B. In Figure 1C, brackets should indicate exactly which bar is compared which bar, not as a vertical line across three bars, which is confusing.

All indicated points are now amended.